# IMPROVING SPARSE-VIEW 3DGS GENERALIZATION VIA FLAT MINIMA OPTIMIZATION

## ABSTRACT

Recent advances in neural rendering have established 3D Gaussian Splatting (3DGS) as a highly efficient representation for novel view synthesis, enabling real-time training and rendering with strong fidelity. However, when supervision is limited to a sparse set of input views, 3DGS tends to overfit to the observed images, resulting in poor generalization to unseen viewpoints. We approach this challenge from the perspective of flat minima (FM) optimization, which seeks solutions that remain stable under small parameter perturbations and are thus more robust. Viewing Gaussian parameters as trainable weights, we adapt FM principles to the geometric and dynamic nature of 3DGS by introducing several key techniques. First, we propose a Scale-Adaptive Perturbation (SAP) scheme that scales perturbation magnitude according to each Gaussian's anisotropy, preserving fine details while promoting robustness. Second, we adopt stochastic perturbation where each Gaussian is probabilistically perturbed or left unchanged, allowing perturbations while preventing oversmoothing of scene details. Third, we schedule the perturbation magnitude to increase gradually during training, avoiding excessive noise before Gaussians capture stable structure. Finally, we incorporate periodic reinitialization of non-positional parameters such as scale, rotation, and opacity, and Spherical Harmonics (SH) coefficients. preventing degenerate cases like elongated Gaussians and maintaining well-conditioned primitives throughout optimization. Together, these techniques form a lightweight framework that integrates seamlessly into existing 3DGS pipelines without architectural changes. Experiments on LLFF and Mip-NeRF360 demonstrate that our method consistently improves both quantitative metrics and perceptual quality under sparse-view supervision, producing reconstructions that are sharper, more stable, and better generalized to novel viewpoints.

## 1 INTRODUCTION

Recent advances in neural rendering have been driven by representative methods such as Neural Radiance Fields (NeRF) Mildenhall et al. (2021) and, more recently, 3D Gaussian Splatting (3DGS) Kerbl et al. (2023). While these approaches have achieved remarkable progress in novel view synthesis (NVS), sparse-view scenarios, where only a few input images are available, remain highly challenging. In such settings, models are prone to overfitting to the input views, leading to poor generalization to novel viewpoints.

In this paper, we aim to tackle this through the lens of flat minima (FM) optimization. Originally developed in the context of neural networks, FM optimization improves generalization by encouraging solutions that reside in flatter regions of the loss landscape, where small parameter perturbations do not significantly increase the loss Hochreiter & Schmidhuber (1997); Dinh et al. (2017); Keskar et al. (2016). Inspired by this principle, we reinterpret 3DGS as a supervised learning system where camera poses serve as inputs, rendered images as outputs, and Gaussian parameters play the role of learnable weights. Within this formulation, overfitting in 3DGS corresponds to sharp-minima solutions that are highly sensitive to parameter shifts; resembling the failure modes addressed by FM theory.

Motivated by these insights, we propose to introduce flat minima optimization into the sparse-view 3DGS setting. Specifically, we apply stochastic perturbations to Gaussian parameters during train-

ing, following the spirit of prior FM techniques that encourage robustness through parameter noise injection Li et al. (2022; 2024b). At each iteration, random noise is added to Gaussian parameters, and the model is trained to minimize reconstruction loss under these perturbations, guiding it toward flatter minima and more generalizable solutions.

Beyond this baseline application of flat minima optimization, we further tailor the approach to the specific characteristics of 3DGS. While directly adding perturbations can improve generalization, naïve strategies often suppress fine-grained geometry, leading to underfitting of local structures. To address this, we adapt FM optimization to better align with the geometric properties and training dynamics of 3DGS.

First, we introduce a Scale-Adaptive Perturbation (SAP) strategy that takes into account the scale and anisotropic shape of each Gaussian. Perturbation noise is sampled in proportion to the Gaussian's spatial extent, applying larger perturbations along longer axes and smaller ones along shorter axes. This ensures that perturbations remain meaningful across primitives of varying sizes, preventing destabilization for small Gaussians while still enforcing robustness for larger ones. In particular, this design helps preserve fine-grained scene details while maintaining the regularization benefits of flat minima optimization.

We adopt a stochastic perturbation scheme inspired by prior FM studies Li et al. (2024b), which combine losses from perturbed and unperturbed models to encourage convergence to flatter solutions. Building on this idea, we instead apply SAP in a stochastic manner. That is, for each iteration, each Gaussian is probabilistically perturbed or left unchanged. This avoids the need to render both perturbed and unperturbed models separately, while keeping the training objective simple and lightweight.

In addition, we complement positional perturbations with periodic parameter reinitialization. Scale, rotation, and opacity and Spherical Harmonics (SH) coefficients are reset at fixed intervals while positions and colors are preserved. This prevents degenerate cases such as elongated Gaussians, providing an additional layer of regularization that stabilizes geometry over the course of training.

Overall, our framework is lightweight, requires no architectural modifications, and integrates seamlessly into existing pipelines. Experiments demonstrate that it consistently improves both quantitative metrics and perceptual quality, producing reconstructions that are sharper, more stable, and better generalized to novel viewpoints.

## 2 RELATED WORK

### 2.1 NOVEL VIEW SYNTHESIS

Novel view synthesis has emerged as a fundamental task in 3D vision, aiming to synthesize unseen views of a scene from a set of input images. Recent advances in Neural Radiance Fields (NeRF) Mildenhall et al. (2021); Barron et al. (2021; 2022); Müller et al. (2022) pioneered the use of neural implicit representations for this task, achieving high-quality results but suffer mainly from slow rendering and training due to per-ray MLP evaluations. To overcome this, variants of 3D Gaussian Splatting (3DGS) Kerbl et al. (2023) were introduced as a fast and explicit alternative. By representing scenes as a set of 3D Gaussians and rendering via differentiable rasterization, 3DGS achieves significantly faster training and real-time rendering while maintaining high reconstruction quality. This makes it particularly attractive for practical deployment and real-time applications.

### 2.2 3DGS UNDER SPARSE-VIEW CONDITION

While 3DGS Kerbl et al. (2023) has demonstrated impressive performance in real-time novel-view synthesis under dense supervision, its effectiveness diminishes in sparse-view scenarios, where limited input views make the reconstruction problem highly ill-posed. In such settings, models tend to overfit to observed views, resulting in poor generalization to unseen viewpoints.

To alleviate this, various approaches have been proposed: some use depth cues from pretrained estimators Chung et al. (2024); Li et al. (2024a); Zhu et al. (2024), others incorporate semantic features Liao et al. (2025), while geometric regularization Zhang et al. (2024) and stochastic training strategies Park et al. (2025); Xu et al. (2025) have also been explored. More recently, an analysis

on co-adaptation in 3DGS under sparse-view supervision Chen et al. (2025) introduced stochastic regularizers such as Gaussian dropout and opacity noise to reduce entanglement among Gaussians. In this work, we pursue an alternative direction by introducing flat minima optimization into 3DGS, perturbing Gaussian parameters during training to promote robustness and generalization under sparse-view supervision.

### 2.3 FLAT MINIMA OPTIMIZATION

Flat minima optimization aims to improve the generalization ability of neural networks by finding regions in the loss landscape that remain low under small parameter perturbations. This idea, rooted in early studies Hochreiter & Schmidhuber (1997); Keskar et al. (2016); Dinh et al. (2017), has gained renewed attention through methods like Sharpness-Aware Minimization (SAM) Foret et al. (2020), which minimizes the worst-case loss via adversarial weight perturbations. To compute the perturbation direction, SAM requires an additional forward and backward pass per iteration, nearly doubling the overall computational overhead.

Meanwhile, Random Weight Perturbation (RWP) Bisla et al. (2022) offers a simpler alternative by injecting random noise into weights without extra gradient computation, although its performance was initially lower than SAM. Later works Li et al. (2024b) refined this approach by mixing perturbed and unperturbed losses and adapting noise magnitudes, achieving stronger performance with minimal cost.

Building upon these insights, we incorporate flat minima optimization into 3D Gaussian Splatting, exploring perturbation-based training as a means to improve generalization. While noise has also been used in 3DGS through sampling-based methods such as 3DGS-MCMC Kheradmand et al. (2024), those approaches integrate noise into parameter updates to facilitate exploration, whereas our perturbations are transient and specifically designed to induce flat minima.

## 3 METHOD

### 3.1 PRELIMINARIES

**3D Gaussian Splatting (3DGS)** represents a scene using a set of anisotropic Gaussians, each defined by learnable parameters including position $x_i \in \mathbb{R}^3$, scale $s_i$, rotation $r_i$, color $c_i$, and opacity $o_i$, where $i$ indexes the $i$-th Gaussian. These Gaussians are projected and composited through differentiable rasterization to render photorealistic images from arbitrary viewpoints.

Given a set of training images and corresponding camera parameters, the Gaussian parameters are optimized by minimizing a photometric reconstruction loss:

$$\theta \leftarrow \theta - \eta \cdot \nabla \mathcal{L}(\theta), \tag{1}$$

where $\theta$ encompasses all learnable parameters and $\mathcal{L}(\theta)$ denotes the rendering loss, typically computed as a combination of L1 and SSIM losses between the rendered and corresponding ground-truth images, and $\eta$ is the learning rate.

**Flat Minima (FM) optimization** aims to improve generalization by encouraging the model to converge to regions in the loss landscape where small perturbations do not significantly increase the loss. Here, the representative frameworks are adversarial and stochastic perturbation-based optimization. Sharpness-Aware Minimization (SAM) Foret et al. (2020) is a representative of the former method, raising the adversarial perturbation scheme. It minimizes the worst-case loss within a small neighborhood around the parameters as following:

$$\mathcal{L}_{\text{SAM}}(\theta) = \min_{\theta} \max_{\|\epsilon\|_2 \leq \rho} \mathcal{L}(\theta + \epsilon). \tag{2}$$

This min-max formulation aggressively biases optimization toward flatter regions of the loss landscape, but requires an additional gradient computation, which significantly increases the training cost. Meanwhile, Random Weight Perturbation (RWP), an approach explored in Bisla et al. (2022),

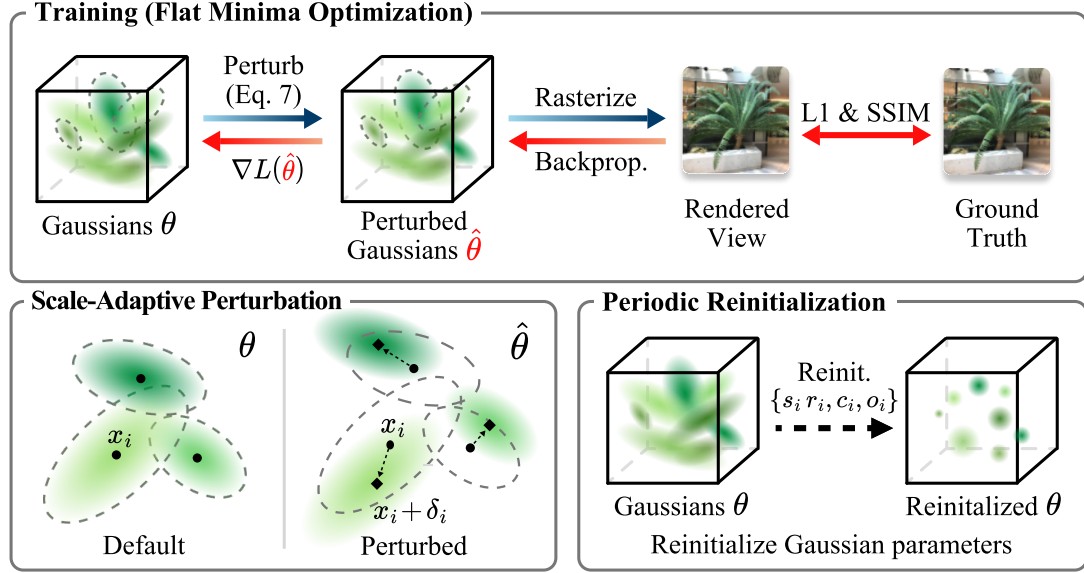

Figure 1: Overview of our flat minima optimization framework for sparse-view 3D Gaussian Splatting (3DGS). Our approach enhances generalization by introducing perturbations to Gaussian positions during training. To respect the geometric structure of the representation, we scale perturbations according to the anisotropic extent of each Gaussian and apply them stochastically at the level of individual Gaussians. In addition, we periodically reinitialize a subset of parameters, including scale, rotation, opacity, and spherical harmonics (SH) coefficients. These techniques encourage flatter minima, improving robustness and reconstruction quality under sparse-view supervision.

provides a more efficient alternative by replacing the worst-case loss with an expected loss over random perturbations:

$$\mathcal{L}_{\text{RWP}}(\theta) = \mathbb{E}_{\epsilon \sim \mathcal{N}(0, \sigma^2 I)} \left[ \mathcal{L}(\theta + \epsilon) \right]. \tag{3}$$

By averaging losses across randomly perturbed weight configurations, RWP effectively smooths the loss landscape without additional backpropagation. Later works Li et al. (2024b) further refined this approach by mixing perturbed and unperturbed losses and adapting noise magnitudes, achieving stronger performance with minimal cost.

In this work, we build upon these insights and adapt flat minima optimization to the context of 3D Gaussian Splatting.

### 3.2 FLAT MINIMA OPTIMIZATION IN 3DGS

We begin by formulating flat minima (FM) optimization in the context of 3D Gaussian Splatting (3DGS). While conventional FM approaches in feed-forward neural networks inject random perturbations into all weights to encourage flatter solutions, directly adopting this strategy in 3DGS overlooks its unique geometric structure. Instead, we reinterpret FM optimization by perturbing the 3D position coordinates $\mathbf{x}_i$ of Gaussians during training. From a geometric perspective, overfitting in sparse-view 3DGS manifests as sharp minima that are highly sensitive to positional shifts, which in turn causes vulnerability in novel view rendering. By introducing position-level perturbations, we guide the optimization toward flatter minima where Gaussian positions are more robust under spatial variation. Accordingly, we extend this idea with perturbation and reinitialization techniques that account for the geometric properties and dynamic behavior of 3DGS.

**Scale-Adaptive Perturbation (SAP).** Prior works on random perturbation strategies Bisla et al. (2022); Li et al. (2024b) have demonstrated the effectiveness of stochastic perturbations in improving generalization. To adapt this principle to the geometric nature of 3DGS, we introduce a Scale-Adaptive Perturbation (SAP) scheme that tailors the perturbation magnitude to the anisotropic scale

of each Gaussian. By aligning perturbations with the spatial extent of each primitive, SAP effectively balances robustness and detail preservation in 3D reconstruction.

Formally, the perturbed parameters $\hat{\theta}$ are defined as

$$\mathbf{x}'_i = \mathbf{x}_i + \delta_i, \quad \delta_i \sim \mathcal{N}\big(0, \gamma^2 \boldsymbol{\Sigma}_i\big), \quad \boldsymbol{\Sigma}_i = \mathbf{R}_i\, \mathbf{S}_i \mathbf{S}_i^\top \mathbf{R}_i^\top, \quad \hat{\theta} = \{\mathbf{x}'_i, s_i, r_i, c_i, o_i\}, \quad (4)$$

where $\mathbf{x}_i$ denotes the 3D position of the $i$-th Gaussian, $s_i$ is its anisotropic scale, $r_i$ its rotation, $c_i$ its color, and $o_i$ its opacity. Here, $\mathbf{R}_i$ is the rotation matrix constructed from $r_i$, and $\mathbf{S}_i = \mathrm{diag}(s_i)$ is the scaling matrix whose diagonal elements correspond to the per-axis scales of the Gaussian. Gradients are then computed at the perturbed parameters $\hat{\theta}$, and the optimization step is performed accordingly:

$$\theta \leftarrow \theta - \eta \cdot \nabla \mathcal{L}(\hat{\theta}). \quad (5)$$

By modulating perturbations according to the anisotropic scale of each Gaussian, SAP enforces stronger regularization on larger or elongated primitives while applying finer noise to smaller ones. This design preserves high-frequency details in compact Gaussians while still promoting robustness across the broader scene geometry.

**Stochastic Application of SAP.** To further adapt flat minima optimization to 3DGS, we introduce stochastic perturbations, where each Gaussian is probabilistically perturbed or left unchanged. This design is motivated by recent Random Weight Perturbation strategies Li et al. (2024b), where perturbed and unperturbed objectives are mixed to encourage flatter solutions. Instead of explicitly rendering two models and combining their losses, we realize a similar effect more efficiently by perturbing each Gaussian with probability $p$ at every iteration. In practice, this avoids the need to perform two forward passes, keeping the training objective simple and lightweight.

Formally, for each Gaussian $i$, the perturbed position $\mathbf{x}'_i$ is defined as

$$\mathbf{x}'_i = \begin{cases} \mathbf{x}_i + \delta_i, & \text{with probability } p, \\ \mathbf{x}_i, & \text{with probability } 1 - p, \end{cases} \quad (6)$$

Applying perturbations in this localized, probabilistic manner offers an additional advantage over global perturbation strategies. In prior FM-inspired methods, where perturbed losses are always incorporated into optimization, we empirically observe a tendency to suppress fine-grained geometry, leading to underfitting of local structures. Similar observations can also be found in our ablation results (see Tab. 5), where the loss-mixing strategy shows degradation in rendering fidelity. By contrast, our per-Gaussian stochastic formulation limits the extent to which perturbations dominate the training signal, allowing the model to retain delicate scene details while still benefiting from the regularization effect. This not only reduces overfitting but also enhances the stability of optimization without incurring additional computational overhead.

**Perturbation Magnitude Scheduling.** We linearly scale the perturbation magnitude from 0 at the beginning to a target value at the final iteration. This gradual increase avoids excessive noise in the early stage, while ensuring that the desired perturbation strength is applied once the model reaches a more stable regime. Empirically, we find that applying strong perturbations too early, before Gaussians have sufficiently captured the scene structure, tends to degrade performance, motivating this progressive schedule.

Formally, the stochastic perturbation is reformulated to incorporate this scheduling scheme, following the linear annealing strategy used in prior work such as Park et al. (2025):

$$\mathbf{x}'_i = \begin{cases} \mathbf{x}_i + \alpha(t)\, \delta_i, & \text{with probability } p, \\ \mathbf{x}_i, & \text{with probability } 1 - p, \end{cases} \quad \text{where} \quad \alpha(t) = \tfrac{t}{T}, \quad (7)$$

where $t$ is the current iteration, $T$ is the total number of iterations, and $\alpha(t)$ linearly increases the perturbation magnitude from 0 to 1 over training.

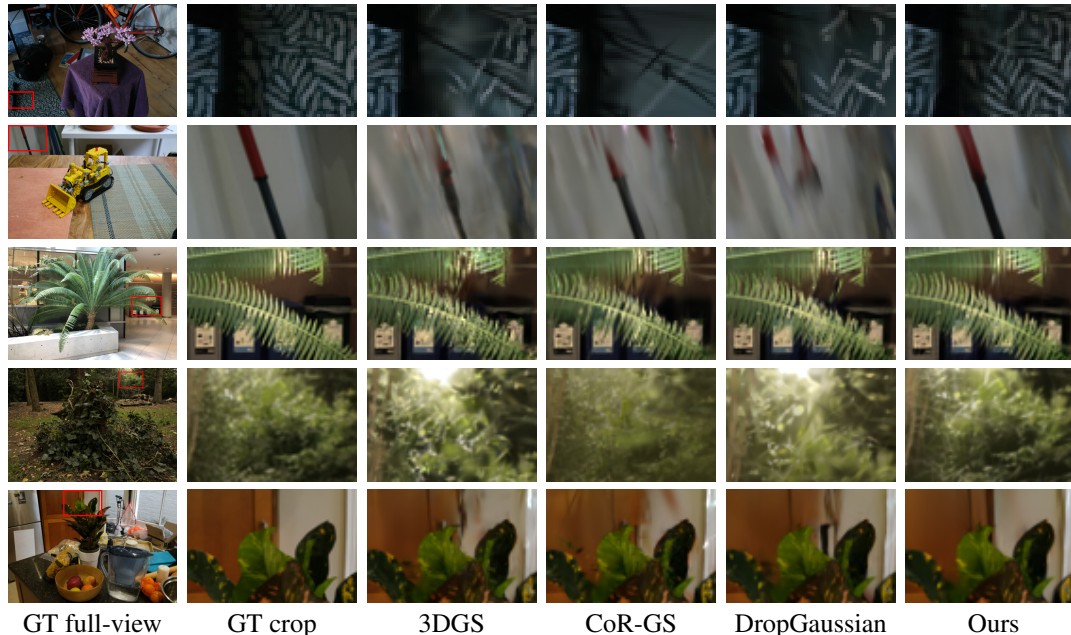

GT full-view     GT crop     3DGS     CoR-GS     DropGaussian     Ours

Figure 2: Qualitative comparison in LLFF and MipNeRF360 dataset. Compared to prior methods, our approach reconstructs sharper details and more geometrically consistent structures, especially in under-constrained regions such as planar surfaces and occlusion boundaries. While baselines often suffer from spatial misalignment or blurring, our method produces visually coherent and high-fidelity results, demonstrating improved generalization to novel viewpoints.

**Periodic Gaussian Reinitialization.** In addition to position-level perturbations, which serve as our main mechanism for flat minima optimization, we periodically reinitialize a subset of Gaussian parameters during training. Specifically, scale, rotation, opacity, and Spherical Harmonics (SH) co-efficients are reset at fixed intervals, and opacity is refreshed following the original opacity reset mechanism in Kerbl et al. (2023), while positions, colors, and the total number of Gaussians are preserved. This design complements position perturbations by preventing degenerate cases such as elongated Gaussians, which positional noise alone is insufficient to address. Note that this reinitialization procedure is similar to the one used when 3DGS is first initialized from a point cloud derived from Structure-from-Motion (SfM) Schonberger & Frahm (2016), where geometric structure is preserved while attributes such as scale, rotation, and appearance are reinitialized. In particular, scales are reset to spherical forms, bringing overly anisotropic Gaussians back into a stable regime where SAP can effectively preserve fine details. By periodically applying this process during training, we enhance stability and further guide optimization toward flatter minima.

## 4 EXPERIMENTS

**Experimental Setup.** We base our implementation on the original 3DGS framework Kerbl et al. (2023), while following most hyperparameters from DropGaussian Park et al. (2025). We set the perturbation probability to $p_{max} = 0.3$, perturbation coefficient $\gamma = 2$, and reinitialize Gaussians every 1,000 iterations. To ensure stability, perturbed positions are clamped so that the displacement does not exceed the corresponding Gaussian's scale. Experiments were conducted on NVIDIA RTX TITAN and A6000 GPU.

**Datasets and Evaluation Metrics.** We evaluate our method on two widely used benchmarks for novel view synthesis: LLFF Mildenhall et al. (2019) and Mip-NeRF360 Barron et al. (2022), both downsampled by a factor of 8. For LLFF, we use 3, 6, and 9 input views, and for Mip-NeRF360, we use 12 and 24 views, following prior works. The reconstruction performance is evaluated by three commonly adopted full-reference image quality assessment (FR-IQA) metrics: peak signal-to-noise ratio (PSNR), structural similarity index (SSIM) Wang et al. (2004), and learned perceptual image

Table 1: Quantitative comparison on the LLFF dataset under 3-view, 6-view, and 9-view settings. Our method achieves consistently strong performance across all metrics, outperforming or matching prior baselines in PSNR, SSIM, and LPIPS. The results demonstrate the effectiveness of our flat-minima optimization framework in improving both photometric accuracy and perceptual quality.

| Method | 3-view | | | 6-view | | | 9-view | | |
|---|---|---|---|---|---|---|---|---|---|
| | PSNR↑ | SSIM↑ | LPIPS↓ | PSNR↑ | SSIM↑ | LPIPS↓ | PSNR↑ | SSIM↑ | LPIPS↓ |
| 3DGS | 19.22 | 0.649 | 0.229 | 23.80 | 0.814 | 0.125 | 25.44 | 0.860 | 0.096 |
| DNGaussian | 19.12 | 0.591 | 0.294 | 22.18 | 0.755 | 0.198 | 23.17 | 0.788 | 0.180 |
| FSGS | 20.43 | 0.682 | 0.248 | 24.09 | 0.823 | 0.145 | 25.31 | 0.860 | 0.122 |
| CoR-GS | 20.45 | 0.712 | 0.196 | 24.49 | 0.837 | 0.115 | 26.06 | 0.874 | 0.089 |
| DropGaussian | 20.76 | 0.713 | 0.200 | 24.74 | 0.837 | 0.117 | 26.21 | 0.874 | 0.088 |
| **Ours** | 20.88 | 0.731 | 0.184 | 24.76 | 0.840 | 0.114 | 26.23 | 0.875 | 0.088 |

Table 2: Quantitative comparison on MipNeRF360 dataset under 12-view and 24-view settings.

| Method | 12-view | | | 24-view | | |
|---|---|---|---|---|---|---|
| | PSNR↑ | SSIM↑ | LPIPS↓ | PSNR↑ | SSIM↑ | LPIPS↓ |
| 3DGS | 18.52 | 0.523 | 0.415 | 22.80 | 0.708 | 0.276 |
| FSGS | 18.80 | 0.531 | 0.418 | 23.70 | 0.745 | 0.230 |
| CoR-GS | 19.52 | 0.558 | 0.418 | 23.39 | 0.727 | 0.271 |
| DropGaussian | 19.74 | 0.577 | 0.364 | 24.13 | 0.762 | 0.225 |
| **Ours** | 19.50 | 0.584 | 0.348 | 24.19 | 0.771 | 0.219 |

Table 3: Quantitative evaluation on LLFF dataset under 3-view setting, with different noise distributions. † indicates our default setting.

| Method | PSNR | SSIM | LPIPS |
|---|---|---|---|
| Anisotropic† | 20.88 | 0.731 | 0.184 |
| Isotropic (max) | 20.73 | 0.725 | 0.188 |
| Isotropic (mean) | 20.67 | 0.724 | 0.185 |
| Isotropic (fixed) | 20.54 | 0.714 | 0.188 |

patch similarity (LPIPS) Zhang et al. (2018). Together, these metrics provide a comprehensive evaluation of both photometric accuracy and perceptual quality of novel view synthesis.

**Baselines.** We compare our method with recent approaches for sparse-view novel view synthesis, including 3DGS Kerbl et al. (2023), CoR-GS Zhang et al. (2024), and DropGaussian Park et al. (2025). We also include DNGaussian Li et al. (2024a) and FSGS Zhu et al. (2024), which incorporate additional geometric supervision using depth priors.

### 4.1 QUANTITATIVE EVALUATION

Tab 1 and Tab 2 summarize the quantitative results across all sparse-view benchmarks. Our method outperforms prior approaches, achieving state-of-the-art performance across datasets and view settings. These results demonstrate the effectiveness of our flat minima framework in enhancing both reconstruction fidelity and generalization under sparse-view supervision.

### 4.2 QUALITATIVE EVALUATION

Fig. 2 presents qualitative comparisons across various sparse-view scenarios. Compared to existing baselines, our method consistently generates sharper and more structurally coherent renderings, particularly under challenging geometric conditions. Baseline methods often suffer from noticeable blurring in background regions where supervision is scarce. These methods also tend to exhibit structural inconsistencies, such as spatial misalignment or distorted geometry. For instance, CoR-GS and DropGaussian occasionally fail to maintain consistent geometry, leading to spatial shifts in novel viewpoints, or inaccurately rendering planar surfaces and straight lines (e.g., second row). Such artifacts indicate insufficient regularization or overfitting to the limited input views.

In contrast, our method produces clean and sharper reconstructions across all scenes. The recovered geometry remains consistent even under significant viewpoint changes, preserving fine-grained details such as object boundaries and linear structures. These improvements highlight the effective-

Table 4: Quantitative evaluation on LLFF dataset under 3-view setting, by varying which Gaussian parameter is perturbed during training. † indicates our default setting.

| Method | PSNR | SSIM | LPIPS |
|---|---|---|---|
| Position† | 20.88 | 0.731 | 0.184 |
| Rotation | 20.23 | 0.710 | 0.195 |
| Scale | 20.22 | 0.707 | 0.195 |
| Opacity | 20.21 | 0.709 | 0.194 |
| Position + Scale | 20.33 | 0.719 | 0.199 |

Table 5: Ablation study comparing stochastic perturbation with global loss mixing on LLFF dataset under the 3-view setting. † indicates our default setting.

| Method | PSNR | SSIM | LPIPS |
|---|---|---|---|
| Stochastic † | 20.88 | 0.731 | 0.184 |
| Mixed Loss | 20.84 | 0.714 | 0.212 |

Table 6: Ablation on Gaussian Reinitialization on LLFF dataset under the 3-view setting. † indicates our default setting.

| Method | PSNR | SSIM | LPIPS |
|---|---|---|---|
| w/ Reinitialization† | 20.88 | 0.731 | 0.184 |
| w/o Reinitialization | 20.58 | 0.713 | 0.197 |

Table 7: Ablation on Perturbation Magnitude Scheduling on LLFF dataset under the 3-view setting. † indicates our default setting.

| Method | PSNR | SSIM | LPIPS |
|---|---|---|---|
| w/ Scheduling† | 20.88 | 0.731 | 0.184 |
| w/o Scheduling | 20.69 | 0.721 | 0.191 |

ness of our flat minima framework in promoting generalizable geometry, and its robustness under sparse-view supervision.

## 4.3 ABLATION STUDIES

**Perturbation Design.** In Tab. 3, we evaluate different strategies for injecting perturbations into Gaussian positions. Our default configuration leverages anisotropic noise that is proportional to the scale of Gaussian axes, promoting both detail preservation and robustness. For comparison, we test isotropic variants where perturbations are scaled by the longest axis of each Gaussian, by the mean axis length, or fixed uniformly across all Gaussians at half the size threshold for determining clone/split targets. Among these, anisotropic perturbation achieves the best balance between fidelity and regularization, while isotropic variants either oversmooth fine structures or under-regularize large primitives. These results confirm the importance of accounting for the geometric shape of each Gaussian when designing perturbations. Nevertheless, we find that any perturbation strategy yields better generalization than the vanilla 3DGS baseline, highlighting that introducing perturbation itself acts as a strong form of regularization, even when the design is not optimal.

**Effect of Perturbing Different Parameters.** To assess the sensitivity of different Gaussian attributes, we applied perturbations individually to position, scale, rotation, and opacity, as reported in Tab. 4. For scale, noise was added proportional to each axis length, following our SAP strategy. Rotation was perturbed by altering each Gaussian's orientation with small random angular offsets around the three principal axes. Opacity was perturbed by adding Gaussian noise directly to its values.

The results indicate that perturbing position is the most effective for improving generalization, consistently outperforming other parameter choices. Interestingly, applying perturbations to both position and scale does not further enhance performance; instead, its performance degraded compared to perturbing position alone. These observations suggest that inducing flatness in the position space is particularly impactful for promoting stable and generalizable geometry in 3DGS, and that geometry-based perturbation plays an important role in our framework, while other parameter groups may still be beneficial under different schedules or formulations and thus remain worth exploring in future work.

**Stochastic Perturbation Strategy.** We further investigate the role of stochasticity in perturbation design. Prior flat minima studies Li et al. (2024b) improve generalization by mixing losses from

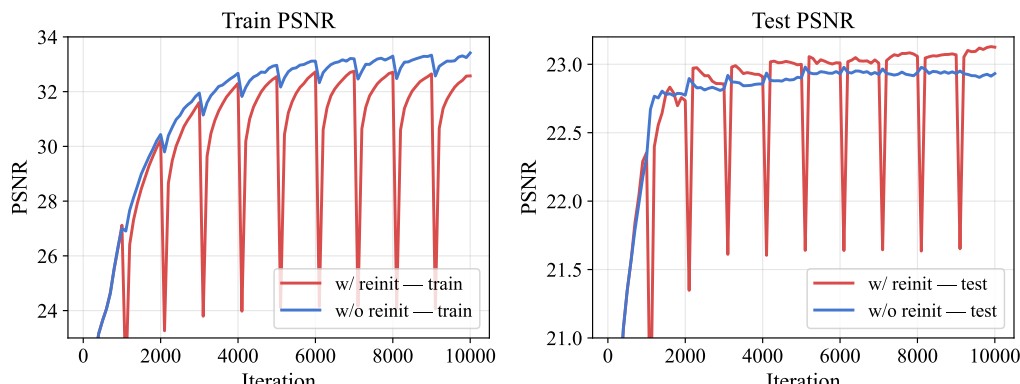

Figure 3: Training and test PSNR trajectories with and without Gaussian reinitialization on the LLFF fern scene under the 3-view setting. Without reinitialization, training PSNR keeps rising steadily while test PSNR plateaus, indicating overfitting. In contrast, periodic reinitialization moderates the growth of training PSNR but enables continuous improvements in test PSNR, suggesting better generalization. The sharper drops observed at each reset interval are due to reinitializing more diverse parameters, but performance quickly recovers and ultimately surpasses the non-reinitialized baseline. This aligns with the role of reinitialization in preventing degenerate Gaussians and maintaining a stable regime where perturbations remain effective.

perturbed and unperturbed models, but this requires rendering twice per iteration. Instead, we realize a similar effect by applying perturbations stochastically during training. Concretely, each Gaussian is randomly perturbed with some probability at every iteration, which introduces variability into the optimization while keeping the training objective simple and lightweight. We found that directly relying on perturbed losses at every step tends to oversmooth high-frequency details, whereas our stochastic scheme alleviates this issue and preserves fine structures. This can also be observed in our ablation results (Tab. 5).

**Perturbation Magnitude Scheduling.** We further validate the role of perturbation scheduling by comparing models trained with and without the progressive scaling of perturbation magnitude (Tab. 7). Without scheduling, strong perturbations are applied from the very beginning of training, when Gaussians have not yet captured the coarse scene structure. As a result, optimization is hindered, leading to unstable convergence and degraded reconstruction quality. In contrast, our scheduling strategy gradually increases perturbation strength as training progresses, allowing Gaussians to first establish a stable representation of the scene before stronger perturbations are introduced. This results in consistently better quantitative performance, supporting our intuition that early excessive noise is detrimental, while a progressive schedule provides a more effective balance between regularization and fidelity.

**Periodic Gaussian Reinitialization.** To validate the effect of our reinitialization strategy, we conducted an ablation study on the LLFF dataset under the 3-view setting (Tab. 6, Fig. 3). When reinitialization is disabled, the additional periodic reset of scale, rotation, and SH coefficients is deactivated, while opacity continues to follow the original reset mechanism introduced in Kerbl et al. (2023). Under this setting, performance drops across all metrics, with PSNR decreasing by 0.3. This indicates that augmenting the standard opacity reset with periodic reinitialization of scale, rotation, and SH coefficients helps avoid degenerate cases such as overly elongated Gaussians, providing a mild stabilizing effect during training. It serves as a complementary mechanism alongside position perturbation, jointly maintaining well-conditioned primitives over time.

## 5 CONCLUSION

We addressed the challenge of sparse-view generalization in 3D Gaussian Splatting (3DGS) through the lens of flat minima optimization. Viewing overfitting as convergence to sharp, position-sensitive solutions, we introduced a set of lightweight strategies that adapt FM principles to 3DGS: Scale-Adaptive Perturbation (SAP) to align noise with Gaussian anisotropy, stochastic application to balance perturbed and unperturbed objectives, perturbation scheduling to stabilize early training, and periodic reinitialization to avoid degenerate primitives. Together, these techniques improve stability, preserve fine details, and enhance generalization without architectural changes, consistently achieving sharper and more robust reconstructions on LLFF and Mip-NeRF360. Our findings highlight flat-minima optimization as a lightweight addition that improves the optimization of 3DGS pipeline without requiring architectural changes or external priors, and also leave open the possibility that alternative perturbation strategies applied to other Gaussian parameters may further improve reconstruction quality in future work.

**Repoducilbility Statement.** All critical implementation details and quantitative results are described in the main paper, enabling the research community to readily reproduce our findings. The datasets employed in our experiments are openly accessible, and we plan to release the source code of our method once the paper is accepted.

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

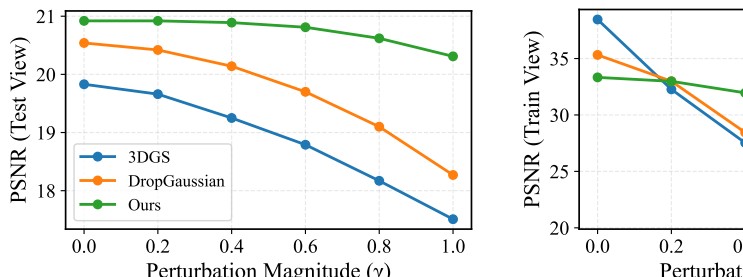

Figure 4: Perturbation robustness analysis on LLFF dataset under 3-view setting. Our method shows smaller performance degradation across perturbation magnitudes compared to 3DGS Kerbl et al. (2023) and DropGaussian Park et al. (2025).

## 6 APPENDIX

**Large Language Model Use.** We used a large language model as a writing and editing assistant and as a tool to search related works. Specifically, it was employed to improve grammar, clarity, and consistency of the manuscript, to reformat LaTeX content such as tables and figures, and to assist with writing code used in the research.

### 6.1 PERTURBATION ROBUSTNESS COMPARISON

In line with prior works that evaluate the sharpness of minima through a model's sensitivity to parameter perturbations (Keskar et al., 2016; Dinh et al., 2017; Foret et al., 2020), we examine how the reconstruction quality of a trained 3DGS model changes when its Gaussian positions are perturbed after optimization. We apply position perturbations using our Scale-Adaptive Perturbation (SAP) with magnitudes $\gamma \in \{0.0, 0.2, 0.4, 0.6, 0.8, 1.0\}$ and evaluate the resulting models on both training and test views.

As shown in Fig. 4, our method shows substantially smaller degradation than both 3DGS Kerbl et al. (2023) and DropGaussian Park et al. (2025) across all perturbation magnitudes. On test views, DropGaussian nearly matches the degradation trend of vanilla 3DGS, while our method shows substantially smaller drops at every scale. The difference is more evident in training views as vanilla 3DGS and DropGaussian exhibit larger degradation, indicating a stronger tendency to overfit to training images compared to our method. This suggests that our optimization strategy leads to flatter and more robust minina than existing 3DGS baselines.

