# OpenReview forum: "Improving Sparse-View 3DGS Generalization via Flat Minima Optimization"
_ICLR.cc/2026/Conference — Submitted to ICLR 2026_

### Official Review · Reviewer_Hzaj · 2025-10-23

**Soundness:** 2
**Presentation:** 1
**Contribution:** 2
**Rating:** 2
**Confidence:** 4

**Summary:**

The paper introduces a flat-minima optimization framework for sparse-view 3DGS that injects scale-adaptive perturbations into Gaussian positions during training to reduce overfitting. It further improves novel view rendering qualities by mixing perturbed and unperturbed objectives, perturbation magnitude scheduling, and periodic re-initialization. It significantly improved the rendering quality of vanilla 3DGS under sparse-view settings.

**Strengths:**

- The integration of Flat Minima Optimization into vanilla 3DGS under sparse view settings to avoid overfitting is well motivated.
- The method is clearly explained and easy to reproduce.
- The ablation studies clearly demonstrate the effectiveness of each of the proposed components.

**Weaknesses:**

- Limited quantitative gain: the main issue of this paper is that it's not convincing that the proposed method really outperforms the baseline method DropGaussian. In most experiments, it only achieves very marginal quantitative improvements. In some occations (e.g., 12-view NVS), the PSNR of DropGaussian is even higher.
- Missing baselines: two strong baselines for sparse-view NVS are missing: MAtCha Gaussians (CVPR 2025) and Difix3D+ (CVPR 2025). They leverage monocular depth priors and diffusion priors to achieve high-quality NVS results, respectively. Comparing to these additional two baselines would provide a more comprehensive assessment of the effectiveness of the proposed method.

**Questions:**

I recommend the authors to add qualitative/quantitative comparisons to the missing baselines MAtCha Gaussians and Difix3D+. I also recommend the authors to provide a side-by-side video comparison with all baselines to help better assess the rendering qualities.

Apart from the mentioned weaknesses, I have an additional question:
- Can the proposed method (at least the probablistic Scale-Adaptive Perturbation part) be implemented as a general plug-in applicable to a wide range of sparse-view Gaussian Splatting pipelines? For instance, if incorporated into baselines such as DropGaussian, would it achieve a comparable relative quantitative improvement over those methods as it does when applied to vanilla 3DGS? Demonstrating consistent relative gains across diverse pipelines would make the contribution significantly more convincing, highlighting the method’s plug-and-play versatility and broad applicability.

I would consider raising my score if the author could add the missing baselines and show a consistent improvement by plugging in the proposed approach into more baseline methods.

---

> ### Author Response · Authors · 2025-12-03
>
> We sincerely appreciate your comment regarding existing sparse-view baselines and potential applicability of our method as a general plug-in module.
>
> ---
>
> ### **Relation to Existing Baselines and Broader Plug-in Applicability**
>
> We appreciate your suggestion to also consider strong prior-based pipelines such as MAtCha Gaussians and Difix3D+. These methods make use of pretrained monocular depth or diffusion models, placing them in a different regime from our work. Our approach focuses on modifying the optimization behavior of 3DGS itself, rather than introducing additional priors. While we believe this difference makes a direct comparison less suitable, we fully agree that it is still important to examine whether our optimization approach can serve as a useful plug-in within such pipelines.
>
> Motivated by this point, we incorporated our method into Difix3D by plugging in our Scale-Adaptive Perturbation into its underlying 3DGS optimization. Even within this stronger prior-guided framework, the integration yields additional improvements. We believe this demonstrates that our method provides a complementary direction that can enhance both prior-free 3DGS baselines and pipelines supported by external depth or diffusion cues.
>
> Please refer to our official author comment for additional quantitative results and discussion.

---

### Official Review · Reviewer_B2SA · 2025-10-28

**Soundness:** 2
**Presentation:** 2
**Contribution:** 2
**Rating:** 2
**Confidence:** 4

**Summary:**

This paper addresses the limited generalization of 3D Gaussian Splatting (3DGS) under sparse-view supervision.
 The authors reinterpret 3DGS optimization as a form of weight learning and propose a set of flat-minima-inspired perturbation strategies to promote robustness.
 Specifically, the method perturbs Gaussian parameters with magnitudes scaled by their anisotropy, applies stochastic perturbations to a random subset of Gaussians, schedules the perturbation strength to increase during training, and periodically reinitializes non-positional parameters to avoid degeneracy.
 Experiments on LLFF and Mip-NeRF360 show consistent improvements over existing baselines such as DropGaussian and CoR-GS.

**Strengths:**

1. Framing sparse-view overfitting as a sharp-minima problem provides a fresh and intuitive perspective.
2. The approach is simple to implement, adds minimal computational cost, and integrates easily into existing 3DGS pipelines.
3. Ablation studies and comparisons are clearly presented, isolating the contributions of each component.

**Weaknesses:**

1. Despite the flat-minima motivation, the method essentially performs Gaussian noise injection—a well-known regularization technique. Similar stochastic or Bayesian formulations (e.g., 3DGS-MCMC) are not cited or compared.
2. No analysis (e.g., curvature or sharpness metrics) is provided to demonstrate that the optimization indeed converges to flatter minima.
3. Reported improvements are small and may fall within expected training variance, raising questions about the actual impact of the proposed components.

**Questions:**

1. Can you provide evidence that the proposed method achieves flatter minima, such as curvature or Hessian-based analysis?
2. How sensitive is performance to perturbation probability, noise scale, and reinitialization frequency?

---

> ### Author Response · Authors · 2025-12-03
>
> We thank you for the constructive and detailed feedback. Your comments have helped us more clearly distinguish our approach from related stochastic methods and to better articulate the flat minima optimization perspective underlying our formulation.
>
> ---
>
> ### **Clarifying the Distinction Between Our Method and 3DGS-MCMC**
>
> We thank you for the helpful comment. We agree that injecting noise can appear similar to a generic regularization technique at first glance, and we appreciate the opportunity to clarify how our method differs from stochastic update formulations in 3DGS, particularly 3DGS-MCMC. In the revised paper, we added the citation and discussion to position our method more clearly with respect to this line of work.
>
> A key distinction is where and how the stochasticity is introduced during optimization. In sampling-oriented formulations such as 3DGS-MCMC, noise is injected directly into the parameter update, which can be formulated as
>
> $θ_{t+1} = θ_t − η ∇L(θ_t) + ε_t$,
>
> so the optimization trajectory itself becomes explicitly stochastic and the perturbations accumulate over time. This accumulated stochasticity primarily supports exploration and makes training less sensitive to initialization.
> In contrast, our method follows a flat minima optimization perspective and applies perturbations only during loss evaluation while keeping the original parameters unchanged. Specifically, we form a perturbed parameter set
>
> $\hatθ_t = θ_t + ε_t$,
>
> evaluate the loss at $\hatθ_t$, and update the unperturbed parameters via
>
> $θ_{t+1} = θ_t − η ∇L(\hatθ_t)$.
>
> With this design, stochasticity does not accumulate along the parameter trajectory. Instead, it locally smooths the loss landscape and encourages solutions that are less sensitive to parameter changes. This non-accumulating perturbation also allows for the use of larger perturbation magnitudes for guiding the optimization toward flatter minima, without causing instability.
> Beyond this fundamental difference, we further adapt the flat minima objective to the training dynamics of 3DGS. Our main design is Scale-Adaptive Perturbation (SAP), which injects noise in proportion to each Gaussian’s anisotropic spatial extent so that the regularization remains meaningful across primitives of different sizes while preserving fine-scale structures. To make this formulation practical for 3DGS, we implement SAP in a per-Gaussian stochastic manner, where each Gaussian is probabilistically perturbed or left unchanged within a single rendering pass. This design is motivated by a prior two-pass scheme that leverages both perturbed and unperturbed objectives, and it reduces the rendering overhead by roughly half while retaining the intended benefits of such mixed objective training.

---

### Official Review · Reviewer_2A7R · 2025-11-04

**Soundness:** 3
**Presentation:** 3
**Contribution:** 2
**Rating:** 4
**Confidence:** 4

**Summary:**

This paper tackles sparse-view 3D Gaussian Splatting (3DGS) overfitting by importing flat-minima (FM) optimization ideas and tailoring them to the geometry of Gaussians. The core method (1) perturbs Gaussian positions with a Scale-Adaptive Perturbation (SAP) whose noise matches each Gaussian’s anisotropy and size, (2) applies those perturbations stochastically (some Gaussians may have perturbations and some may not) per-Gaussian to avoid oversmoothing, (3) linearly schedules perturbation strength over training, and (4) periodically reinitializes non-positional parameters (scale, rotation, opacity, SH) to prevent degenerate elongated Gaussians. Notably, the method is lightweight and does not require any architectural changes to 3DGS. The method improves PSNR/SSIM and lowers LPIPS versus baselines. Ablations confirm position noise with anisotropic scaling, stochastic application, scheduling, and reinitialization each contribute to the gains. Qualitatively, results show sharper details and better geometric consistency in under-constrained regions.

**Strengths:**

1. Well-motivated approach: The connection between flat minima optimization and sparse-view generalization in 3DGS is intuitive and well-articulated. Viewing overfitting as convergence to sharp minima is a reasonable perspective.
2. Comprehensive ablations: The paper includes thorough ablation studies examining each component (perturbation design, parameter choices, stochastic strategy, scheduling, reinitialization), providing evidence for design decisions.
3. Practical and lightweight: The method integrates seamlessly into existing 3DGS pipelines without architectural changes or significant computational overhead.
4. Consistent improvements: Results show steady gains across multiple datasets, view settings, and metrics (PSNR, SSIM, LPIPS), demonstrating robustness.

**Weaknesses:**

1. The core ideas (perturbation-based optimization, parameter reinitialization) are well-established techniques. While the adaptation to 3DGS is reasonable, the conceptual contribution feels incremental.
2. While consistent, the quantitative gains are relatively small. Some qualitative differences (eg. in Figure 2) are subtle.
3. The paper doesn't provide rigorous analysis of why position perturbations specifically lead to flatter minima or how the proposed techniques relate to formal FM theory. No measurement of actual loss landscape sharpness before/after.
4. The method introduces several hyperparameters. Limited discussion of sensitivity to these choices or guidance on setting them for new scenarios.

**Questions:**

1. How does computational cost compare to baselines? While you mention it's lightweight, actual training time and memory comparisons would be helpful.
2. Why perturb only positions and not jointly optimize with other parameters? Table 4 shows position is best, but have you tried learned perturbation schedules for different parameter types?
3. Can this approach be combined with other sparse-view methods (depth priors, semantic features) for further improvements?
4. What happens in extremely sparse settings (e.g., 2 views)? Are there failure cases where your method doesn't help?


Additionally, Figure 1 doesnt seem to be rendering on PDF viewers.

---

> ### Author Response · Authors · 2025-12-03
>
> We thank you for carefully reading our paper and for providing insightful comments. Your feedback has helped us sharpen both the positioning and the evaluation of our approach.
>
> ---
>
> ### **Conceptual Contribution Beyond Established Techniques**
>
> We agree that perturbation-based optimization and parameter reinitialization are established ideas, and we appreciate the opportunity to clarify our conceptual contribution beyond applying these techniques in a generic way.
>
> Our main contribution is to bring a flat minima (FM) optimization perspective to sparse-view 3DGS by explicitly viewing 3DGS training as a supervised learning problem where Gaussian parameters function as learnable weights. Under this viewpoint, sparse-view overfitting corresponds to solutions that are highly sensitive to small parameter shifts, aligning with the sharp-minima failure mode discussed in FM theory.
>
> Building on this perspective, our method adapts FM optimization to the geometric structure and dynamics of 3DGS rather than injecting noise in an undifferentiated manner. We design the perturbations to be scale-adaptive to each Gaussian’s anisotropic extent, apply them in a lightweight per-Gaussian manner within a single rendering pass, and pair them with periodic reinitialization of non-positional parameters to avoid degeneracies during sparse-view training.
>
> Also, prior FM study implemented perturbed and unperturbed objectives through two separate forward passes to aid convergence. In contrast, we obtain a similar effect within a single rendering pass through per-Gaussian stochastic perturbation, offering a more efficient instantiation of FM-style training that we believe provides a useful addition to both FM optimization and sparse-view 3DGS.
>
> ---
>
> ### **Computational Cost Analysis**
>
> We thank you for raising the important question regarding computational overhead. To provide a clear comparison, we report training time, memory usage, and per-Gaussian normalized metrics on the 'fern' scene in LLFF dataset (3-view). In addition to 3DGS and DropGaussian, we include FSGS as a representative depth-prior method, to compare the computational costs relative to prior-guided pipelines.
>
> | Metrics | 3DGS | DropGaussian | Ours | FSGS |
> | --- | --- | --- | --- | --- |
> | Total Time (s) | 122.45 | 116.57 | 150.49 | 1263.71 |
> | Peak Mem (GB) | 0.211 | 0.201 | 0.268 | 2.386 |
> | Gaussians | 69,134 | 60,642 | 76,045 | 127,143 |
> | Avg. Iter Time (ms) / Gaussians | $1.80 \times 10^{-4}$ | $1.87 \times 10^{-4}$ | $2.31 \times 10^{-4}$ | $1.04 \times 10^{-3}$ |
> | Peak Mem (GB) / Gaussians | $3.05 \times 10^{-6}$ | $3.31 \times 10^{-6}$ | $3.52 \times 10^{-6}$ | $1.88 \times 10^{-5}$ |
>
> As the table shows, our method retains training time and memory usage close to those of vanilla 3DGS. DropGaussian appears slightly faster, which we assume is due to its dropout mechanism reducing the number of Gaussians receiving gradient updates.
>
> By contrast, FSGS relies on an external depth network and requires substantially more computation. In per-Gaussian terms, our method is approximately 4× faster and uses about 5× less memory. This efficiency gap becomes even larger when considering total computations: the total training time of FSGS exceeds 1200 seconds, making our method roughly 8× faster, and its peak memory usage is about 8× higher. Taken together, these comparisons indicate that our FM optimization offers an efficient and lightweight path to improved generalization without the computational burden associated with external priors.

---

> > ### Author Response · Authors · 2025-12-03
> >
> > ### **Extremely Sparse Settings and Failure Cases**
> >
> > We conducted an additional 2-view experiment on the LLFF dataset, using COLMAP initialization obtained from the 3-view setting. As expected, the 2-view setting is substantially more challenging, and we observe that PSNR drops by more than 1.5 on average compared to the 3-view setting.
> >
> > The results below show that our method clearly improves over vanilla 3DGS in the 2-view setting. Compared to DropGaussian, our method achieves comparable PSNR while yielding slightly better SSIM and LPIPS, which we attribute to SAP preserving fine-scale structures more effectively. Moreover, combining DropGaussian with our method provides further improvements in PSNR and SSIM.
> >
> > | Method | PSNR | SSIM | LPIPS |
> > | --- | --- | --- | --- |
> > | 3DGS | 18.52 | 0.612 | 0.252 |
> > | DropGaussian | 19.22 | 0.649 | 0.237 |
> > | Ours | 19.24 | 0.656 | 0.227 |
> > | DropGaussian + Ours | 19.49 | 0.664 | 0.231 |
> >
> > Regarding failure cases, extremely sparse views can induce large-scale artifacts and severe geometry ambiguity that our approach may not fully address, since the influence of Scale-Adaptive Perturbation is inherently bounded by each Gaussian’s spatial extent. Consistent with this, the performance margin over vanilla 3DGS is smaller in the 2-view setting than in the 3-view case.
> >
> > ---
> >
> >
> > ### **PDF Rendering Issue**
> >
> > Thank you for pointing out the rendering issue with Figure 1. We also confirmed that the figure fails to display on certain devices. In the revised paper, we have updated Figure 1 to ensure correct rendering across all environments.

---

### Official Review · Reviewer_v3C8 · 2025-11-11

**Soundness:** 3
**Presentation:** 3
**Contribution:** 3
**Rating:** 8
**Confidence:** 4

**Summary:**

This paper presents a novel optimization framework for improving the generalization of 3D Gaussian Splatting (3DGS) in sparse-view novel view synthesis. Traditional 3DGS methods tend to overfit when trained on limited input images, leading to poor reconstruction quality in unseen viewpoints. The authors propose incorporating Flat Minima (FM) optimization—a concept from neural network training—into 3DGS to mitigate this issue. Specifically, they introduce a Scale-Adaptive Perturbation (SAP) scheme that applies probabilistic, geometry-aware perturbations to Gaussian positions, along with perturbation scheduling and periodic parameter reinitialization to improve robustness and prevent overfitting. Experiments on LLFF and Mip-NeRF360 datasets demonstrate that this method achieves consistent improvements in PSNR, SSIM, and LPIPS compared to existing 3DGS baselines (DropGaussian, CoR-GS, and DNGaussian). Ablation studies validate the importance of each proposed module, showing that the FM-inspired perturbations improve both fidelity and generalization, particularly under sparse-view settings.

**Strengths:**

1. While this paper builds on existing FM optimization techniques for improving model generalization, extending parameter perturbation from traditional neural network weight training to gaussian splatting field fitting is entirely novel. The Scale-Adaptive Perturbation method, in particular, represents a unique application of perturbation.

2. The paper clearly outlines each design decision in its perturbation scheme, with every section providing information relevant to understanding the methodology and results.

3. The paper offers strong evidence for its conclusions through quantitative tables and rigorous ablative studies that motivate each design decision.

4. The significance to the field of sparse view novel synthesis is clear: the paper delivers a gaussian splatting field fitting pipeline that improves both quantitative and qualitative results while opening a new direction for future methods—the incorporation of FM optimization techniques and ideas.

**Weaknesses:**

1. Regarding presentation, Equation 4 does not clearly show how sampling covariance depends on the scale and rotation of a given Gaussian kernel, though this relationship is illustrated in the Scale Adaptive Perturbation section of Figure 1.
2. Additionally, while the tables covering the ablation studies are informative, including visual ablative results similar to those in Figure 2 would strengthen the argument.
3. In terms of soundness, one aspect requiring further investigation is the scope of experiments on perturbing different parameters. Although evidence suggests that perturbing position alone yields the best overall performance, the experiments and accompanying discussion are insufficient to prove this conclusively. Notably absent are perturbation experiments on elements such as spherical harmonic coefficients. The paper would benefit from expanding these experiments or acknowledging that further exploration may be necessary for future work.

**Questions:**

My questions are primarily related to the concerns I had about your “Effect of Perturbing Different Parameters” section. I feel the section/ablative study indicates that applying perturbation to other parameters of gaussian splatting fields will not improve performance. It is not quite clear if this is a claim you intend to make or if you encourage further exploration in this area.

---

> ### Author Response · Authors · 2025-12-03
>
> We sincerely thank you for the positive evaluation and for the detailed and constructive feedback. Your comments helped us clarify our presentation and better explain the scope and implications of our experiments.
>
> ---
>
> ### **Covariance Notation in Equation 4**
> We appreciate your helpful comment regarding the ambiguity in the original formulation of Equation 4. The previous notation did not clearly convey how the sampling covariance depends on each Gaussian’s anisotropic scale and rotation. In the revised manuscript, we have updated Equation 4 to make this dependency explicit by defining the covariance as
> $Σ_i = R_i S_i S_i^T R_i^T$,
> where $R_i$ is the rotation matrix constructed from $r_i$ and $S_i = diag(s_i)$ denotes the per-axis scaling matrix.
>
> The revised equation is as follows:
>
> $\mathbf{x}_i' = \mathbf{x}_i + \delta_i$,
> $\delta_i \sim \mathcal{N}\!\bigl(0, \gamma^2 \mathbf{\Sigma}_i\bigr)$,
> $\mathbf{\Sigma}_i = \mathbf{R}_i\, \mathbf{S}_i \mathbf{S}_i^{\top} \mathbf{R}_i^{\top}$,
> $\hat{\theta}$ = {$\mathbf{x}_i', s_i, r_i, c_i, o_i$}.
>
> ---
>
> ### **Scope of Perturbation Experiments**
>
> Thank you for the thoughtful question. We do not intend to claim that perturbing other Gaussian parameters is ineffective in general. Our ablations show that under our flat minima optimization, position perturbation yields the most stable gains. In our case, we apply Scale-Adaptive Perturbation that accounts for the anisotropic shape of each 3D Gaussian, ensuring that perturbation magnitudes reflect the underlying geometric structure.
> We expect that other parameter groups may also prove beneficial under different optimization strategies or perturbation formulations, and exploring these directions remains open for future investigation, which we now note in the revised Experiment and Conclusion section. As an example, the public comment highlights that another framework focusing on the co-adaptation problem of Gaussians reported its strongest gains from opacity noise and Gaussian dropout. This further illustrates that the effectiveness of a perturbation is closely tied to the specific optimization scheme in which it operates.

---

### Public Comment · ~Kangjie_Chen2 · 2025-11-12
**Clarification on Overlap With Prior Noise-Injection Studies in 3DGS (Quantifying and Alleviating Co-Adaptation in Sparse-View 3D Gaussian Splatting)**

I would like to note that our NeurIPS 2025 paper “Quantifying and Alleviating Co-Adaptation in Sparse-View 3D Gaussian Splatting” (https://openreview.net/forum?id=GrPo8NTtzK) has already conducted a comprehensive analysis of noise injection applied to Gaussian parameters (opacity, scale, position, SH, etc.) as a means to reduce co-adaptation in 3DGS. The idea explored in this submission is therefore highly related to what we previously studied.

Since the paper currently does not cite this line of research, I suggest adding the reference for proper attribution and to clarify what new insights this submission provides beyond existing work.

---

> ### Author Response · Authors · 2025-12-03
>
> Thank you for your comment and for bringing your work to our attention. We have added the appropriate citation in section 2.2 of revised paper.
>
> While both works involve perturbing Gaussian parameters under sparse-view supervision, the motivations and methodological choices differ. Your study analyzes co-adaptation in 3DGS and evaluates several forms of noise injection, ultimately adopting Gaussian dropout and opacity noise to mitigate entanglement among Gaussians. Our approach is based on flat-minima optimization and focuses on scale-adaptive position perturbations applied only during loss evaluation, which guide the clean parameters toward flatter regions of the loss landscape. Although position perturbation was also considered in your analysis, our formulation differs in that we integrate it into a flat minima optimization framework and design it to be scale-adaptive. This perspective led us to use position perturbation as a central component, and we found it to be effective when applied transiently during loss evaluation to encourage flatter solutions.
>
> We appreciate your comment, which helped us better position our contribution within the related literature.

---

### Author Response · Authors · 2025-12-03
**Official Author Comment (Part 1/2)**

We sincerely thank the reviewers, area chairs, senior area chairs, program chairs, and everyone supporting the review process. We deeply appreciate the time and care invested in reviewing our work under this year’s challenging circumstances.

To provide a unified response to the recurring points raised across the reviews, we summarize here the clarifications most frequently requested. Reviewers emphasized understanding how our method interacts with existing sparse-view baselines and stronger pipelines, how to substantiate the claimed flat-minima behavior, and how sensitive the approach is to its key hyperparameters. Reviewer Hzaj also noted that clearer evidence of broader applicability would motivate a higher score. The following sections address these shared concerns directly. Additional experimental results have been placed in the appendix for convenience, and we will carefully integrate them into the main manuscript for the camera-ready version.

---

### **Plug-in Compatibility Across Diverse Pipelines**
Our method aims to improve the optimization behavior of 3DGS in sparse-view settings by guiding training toward solutions that are more stable under parameter perturbations. We compare against prior-free baselines such as CoR-GS and DropGaussian, and find that our method achieves performance competitive with these approaches. Furthermore, because the method acts directly on the optimization process rather than introducing additional priors or structural constraints, its effect is largely orthogonal to existing regularization or prior-based strategies, allowing it to be incorporated as a plug-in component to diverse 3DGS pipelines.

To further examine the complementary nature of our optimization strategy, we integrate our method into several representative pipelines without modifying their original architectures. This includes both prior-free methods and those relying on additional cues such as monocular depth–based regularization (as suggested by Reviewer 2A7R) and diffusion-prior–based refinement (Reviewer Hzaj). Across these pipelines, incorporating our optimizer yields clear improvements, indicating that the proposed strategy can benefit 3DGS frameworks that rely on different forms of priors and regularization mechanisms.

**LLFF dataset (3-view)**
| Method                | PSNR  | SSIM  | LPIPS |
|-----------------------|-------|-------|-------|
| FSGS                  | 20.52 | 0.700 | 0.204 |
| FSGS + **Ours**           | **21.03** | **0.725** | **0.196** |
| DropGaussian          | 20.76 | 0.713 | 0.200 |
| DropGaussian + **Ours**   | **21.05** | **0.734** | **0.192** |

**Mip-NeRF360 dataset (12-view)**
| Method           | PSNR  | SSIM  | LPIPS |
|------------------|-------|-------|-------|
| Difix3D          | 20.61 | 0.616 | 0.221 |
| Difix3D + **Ours**   | **20.76** | **0.624** | **0.216** |

---

### **Analysis of Convergence to Flatter Minima**

To examine whether our optimization procedure encourages convergence toward flatter regions of the 3DGS loss landscape, we conduct a robustness analysis following common practices in previous flat minima studies. After the training process, we perturb the Gaussian positions using our Scale-Adaptive Perturbation (SAP) with perturbation magnitudes γ ∈ {0.0, 0.2, 0.4, 0.6, 0.8, 1.0} and measure the rendered reconstruction quality. Across all perturbation strengths, our method displays noticeably smaller degradation than 3DGS and DropGaussian. This pattern is especially pronounced on training views, where the baselines exhibit greater sensitivity to overfit and consequently deteriorate more sharply under the same parameter perturbations. Further details are provided in Appendix Section 6.1.

**PSNR drop in Test Views (LLFF dataset 3-view)**
| Method        | γ=0.2   | γ=0.4   | γ=0.6   | γ=0.8   | γ=1.0   |
|---------------|---------|---------|---------|---------|---------|
| 3DGS          | -0.17   | -0.58   | -1.04   | -1.66   | -2.32   |
| DropGaussian  | -0.12   | -0.40   | -0.84   | -1.44   | -2.27   |
| **Ours**          | **0.00**    | **-0.03**   | **-0.11**   | **-0.30**   | **-0.61**   |

**PSNR drop in Train Views (LLFF dataset 3-view)**
| Method        | γ=0.2  | γ=0.4  | γ=0.6  | γ=0.8  | γ=1.0  |
|---------------|--------|--------|--------|--------|--------|
| 3DGS          | -6.19  | -10.93 | -14.15 | -16.09 | -17.69 |
| DropGaussian  | -2.33  | -6.87  | -9.54  | -11.06 | -13.19 |
| **Ours**          | **-0.35**  | **-1.37**  | **-3.06**  | **-5.08**  | **-7.06**  |

---

> ### Author Response · Authors · 2025-12-03
> **Official Author Comment (Part 2/2)**
>
> ### **Hyperparameter Sensitivity**
>
> We evaluate the sensitivity of our method to two key hyperparameters in the LLFF 3-view setting using the average of five independent runs: the perturbation probability $p$ and the reinitialization interval. Sweeping $p$ from 0.1 to 0.7 yields performance within a narrow range, with slightly stronger results around moderate values; our default choice is $p$ = 0.3. For the reinitialization interval, we use 1000 iterations as the default setting. Reinitialization performed too frequently (e.g., every 500 iterations) leads to a noticeable drop in performance, whereas intervals in the range of 1000–3000 behave similarly and do not exhibit substantial differences.
>
> | Metrics | $p$=0.1 | $p$=0.2 | $p$=0.3 | $p$=0.4 | $p$=0.5 | $p$=0.6 | $p$=0.7 |
> |---------|-------|-------|-------|-------|-------|-------|-------|
> | PSNR    | 20.68 | 20.80 | 20.84 | 20.78 | 20.80 | 20.71 | 20.68 |
> | SSIM    | 0.722 | 0.726 | 0.727 | 0.726 | 0.726 | 0.724 | 0.721 |
> | LPIPS   | 0.186 | 0.185 | 0.187 | 0.190 | 0.193 | 0.198 | 0.203 |
>
> | Metrics | reinit. 0.5k | reinit. 1k | reinit. 2k | reinit. 3k |
> |---------|------------|-------------|-------------|-------------|
> | PSNR    | 20.60      | 20.84       | 20.83       | 20.82       |
> | SSIM    | 0.720      | 0.728       | 0.726       | 0.724       |
> | LPIPS   | 0.198      | 0.188       | 0.185       | 0.187       |
>
> Overall, the method shows stable behavior across these hyperparameters, and the trends suggest that our optimization strategy can be applied within existing 3DGS pipelines without requiring delicate tuning.
>
> ---
>
> Finally, we briefly summarize the core contributions of our work in a consolidated form:
> - We introduce a flat-minima–inspired optimization strategy specifically tailored to the geometric and training behavior of 3DGS, treating Gaussian primitives as learnable parameters within a supervised learning framework.
> - The method achieves stable and competitive results under sparse supervision while improving robustness to parameter perturbations and showing stable behavior without delicate hyperparameter tuning.
> - Because our approach operates directly at the optimization level, it integrates naturally into existing 3DGS variants without architectural changes. Experiments further show that incorporating it into both prior-free systems and pipelines leveraging depth or diffusion-based priors provides complementary improvements.
>
> We again sincerely appreciate the thoughtful evaluations and the time and effort invested throughout the review process, and we are grateful to everyone involved for the guidance that has helped us strengthen our work.

---

### Meta-Review · Area_Chair_AqAh · 2026-01-09

**Summary:**

This paper proposes a perturbation-based optimization strategy for sparse-view 3D Gaussian Splatting, motivated by the perspective that overfitting in sparse-view settings corresponds to convergence to sharp minima. By adapting flat-minima-inspired optimization techniques to the Gaussian splatting field fitting process, and introducing scale-adaptive perturbations, the work offers an intuitive and practically lightweight modification to existing 3DGS pipelines. The method is clearly described, easy to reproduce, and supported by extensive ablation studies that carefully motivate individual design choices.

However, despite these positive aspects, AC finds that the overall research contribution and empirical impact are limited, which weakens the case for acceptance.

Conceptually, the core ideas—noise-based parameter perturbation, stochastic regularization, and periodic reinitialization—are well-established techniques in neural network optimization. While extending these ideas to 3D Gaussian Splatting is reasonable and technically sound, the contribution remains largely incremental, amounting to a domain-specific adaptation rather than a fundamentally new optimization principle. Although the paper frames the method through the lens of flat minima optimization, no direct evidence (e.g., curvature, Hessian-based sharpness metrics, or loss landscape analysis) is provided to demonstrate that the proposed approach indeed leads to flatter minima. As a result, the connection to formal flat-minima theory remains intuitive rather than empirically or theoretically substantiated.

From an experimental standpoint, the reported improvements are consistent but modest, and in several cases comparable baselines such as DropGaussian achieve similar or even better performance. Given the small margins of improvement, it is unclear whether the gains exceed expected training variance. Moreover, several strong and highly relevant recent sparse-view NVS methods (e.g., MAtCha Gaussians, Difix3D+) are missing from the comparisons, making it difficult to accurately position the proposed method within the current state of the art.

The experimental analysis of perturbing different Gaussian parameters is also incomplete. While the paper suggests that position perturbation yields the best performance, the absence of thorough experiments on other parameters (e.g., spherical harmonic coefficients) prevents a definitive conclusion. This leaves open whether the proposed perturbation strategy is truly optimal or merely one reasonable choice among many. Additionally, sensitivity to hyperparameters such as perturbation probability, noise scale, and reinitialization frequency is not sufficiently explored, limiting guidance for practitioners.

Finally, although the method is described as lightweight, the paper does not provide concrete training time or memory comparisons to substantiate this claim. Some presentation issues (e.g., clarity of Equation 4, missing visual ablations corresponding to quantitative tables, and minor figure rendering problems) further detract from the overall polish.

In summary, this paper presents a well-motivated and carefully engineered regularization technique for sparse-view 3D Gaussian Splatting, and the idea of importing flat-minima-inspired optimization into this domain is interesting. However, the incremental nature of the contribution, limited quantitative gains, lack of direct evidence supporting the flat-minima hypothesis, and incomplete baseline comparisons significantly weaken its impact. For these reasons, AC recommends Reject, while encouraging the authors to strengthen the empirical validation and theoretical grounding in future work.

**Reviewer Scores:**

They might be unchanged.

---

### Decision · Program_Chairs · 2026-01-26

Reject